# The Future of Origin of Life Research: Bridging Decades-Old Divisions

**DOI:** 10.3390/life10030020

**Published:** 2020-02-26

**Authors:** Martina Preiner, Silke Asche, Sidney Becker, Holly C. Betts, Adrien Boniface, Eloi Camprubi, Kuhan Chandru, Valentina Erastova, Sriram G. Garg, Nozair Khawaja, Gladys Kostyrka, Rainer Machné, Giacomo Moggioli, Kamila B. Muchowska, Sinje Neukirchen, Benedikt Peter, Edith Pichlhöfer, Ádám Radványi, Daniele Rossetto, Annalena Salditt, Nicolas M. Schmelling, Filipa L. Sousa, Fernando D. K. Tria, Dániel Vörös, Joana C. Xavier

**Affiliations:** 1Institute of Molecular Evolution, University of Düsseldorf, 40225 Düsseldorf, Germany; gargs@hhu.de (S.G.G.); tria@hhu.de (F.D.K.T.); 2School of Chemistry, University of Glasgow, Glasgow G128QQ, UK; 2304694A@student.gla.ac.uk; 3Department of Chemistry, University of Cambridge, Lensfield Road, Cambridge CB2 1EW, UK; sb2391@cam.ac.uk; 4School of Earth Sciences, University of Bristol, Bristol BS8 1RL, UK; hb1864@bristol.ac.uk; 5Environmental Microbial Genomics, Laboratoire Ampère, Ecole Centrale de Lyon, Université de Lyon, 69130 Ecully, France; adrien.boniface@univ-lyon1.fr; 6Origins Center, Department of Earth Sciences, Utrecht University, 3584 CB Utrecht, The Netherlands; e.camprubicasas@uu.nl; 7Space Science Center (ANGKASA), Institute of Climate Change, Level 3, Research Complex, National University of Malaysia, UKM Bangi 43600, Selangor, Malaysia; kuhan@ukm.edu.my; 8Department of Physical Chemistry, University of Chemistry and Technology, Prague, Technicka 5, 16628 Prague 6–Dejvice, Czech Republic; 9UK Centre for Astrobiology, School of Chemistry, University of Edinburgh, Edinburgh EH9 3FJ, UK; valentina.erastova@ed.ac.uk; 10Institut für Geologische Wissenschaften, Freie Universität Berlin, 12249 Berlin, Germany; nozair.khawaja@fu-berlin.de; 11Lycée Colbert, BP 50620 59208 Tourcoing Cedex, France; gladys.kostyrka@wanadoo.fr; 12Institute of Synthetic Microbiology, University of Düsseldorf, 40225 Düsseldorf, Germany; machne@hhu.de (R.M.); nicolas.schmelling@hhu.de (N.M.S.); 13Quantitative and Theoretical Biology, University of Düsseldorf, 40225 Düsseldorf, Germany; 14School of Biological and Chemical Sciences, Queen Mary University of London, London E1 4DQ, UK; giacomo.moggioli@qmul.ac.uk; 15Université de Strasbourg, CNRS, ISIS, 8 allée Gaspard Monge, 67000 Strasbourg, France; muchowska@unistra.fr; 16Archaea Biology and Ecogenomics Division, University of Vienna, 1090 Vienna, Austria; sinje.neukirchen@univie.ac.at (S.N.); a00807114@unet.univie.ac.at (E.P.); filipa.sousa@univie.ac.at (F.L.S.); 17Cellular and Molecular Biophysics, Max Planck Institute of Biochemistry, 82152 Martinsried, Germany; peter@biochem.mpg.de; 18Department of Plant Systematics, Ecology and Theoretical Biology, Eötvös Loránd University, Pázmány Péter sétány 1/C, 1117 Budapest, Hungarymrdanielred@gmail.com (D.V.); 19Institute of Evolution, MTA Centre for Ecological Research, Klebelsberg Kuno u. 3., H-8237 Tihany, Hungary; 20Department of Cellular, Computational and Integrative Biology (CIBIO), University of Trento, 38123 Trento, Italy; daniele.rossetto@unitn.it; 21Systems Biophysics, Physics Department, Ludwig-Maximilians-Universität München, 80799 Munich, Germany; a.salditt@physik.uni-muenchen.de; 22Cluster of Excellence on Plant Sciences (CEPLAS), University of Cologne, 50674 Cologne, Germany

**Keywords:** origins of life, prebiotic chemistry, early life, LUCA, abiogenesis, top-down, bottom-up, emergence

## Abstract

Research on the origin of life is highly heterogeneous. After a peculiar historical development, it still includes strongly opposed views which potentially hinder progress. In the 1st Interdisciplinary Origin of Life Meeting, early-career researchers gathered to explore the commonalities between theories and approaches, critical divergence points, and expectations for the future. We find that even though classical approaches and theories—e.g., bottom-up and top-down, RNA world vs. metabolism-first—have been prevalent in origin of life research, they are ceasing to be mutually exclusive and they can and should feed integrating approaches. Here we focus on pressing questions and recent developments that bridge the classical disciplines and approaches, and highlight expectations for future endeavours in origin of life research.

## 1. Introduction

Understanding the origin of life (OoL) is one of the major unsolved scientific problems of the century. It starts with the lack of a commonly accepted definition of the phenomenon of life itself [1], but difficulties go far beyond merely that obstacle. OoL research involves a large number of diffuse concepts cornering several natural sciences and philosophy, such as entropy, information and complexity. Despite evidence that untangling this knot will require a concerted and collaborative effort between different disciplines, technologies, individuals and groups [2], division in OoL research is still marked, concerning both theories (e.g., RNA world vs. metabolism-first) and approaches (e.g., bottom-up vs. top-down). What causes these on-going divisions, and how can heated debates be moderated?

There is some consensus on a few points. First, the earliest undisputed fossil evidence places life on Earth prior to 3.35 Ga [3] and molecular clocks suggest an origin prior to the late heavy bombardment >3.9 Ga [4]. Second, the origin of life must have resulted from a long process or a series of processes, not a sudden event, for the complexity of a cell could not have appeared instantaneously. The OoL must have started from simple abiotic processes, involving one or more sources of energy and matter (particularly CHNOPS: carbon, hydrogen, nitrogen, oxygen, phosphorus and sulfur, the six most prevalent elements in life on Earth) forming protometabolism, compartmentalization and inheritance. But strikingly, the list of agreements does not expand much further than this. Several researchers have speculated on life forms different than the ones we know and see on Earth—not based on cells [5] or even on CHNOPS [6,7]. Even if evidence for those alternative life forms has yet to be found, here we do not exclude their possibility. Throughout this text we will focus on life as we know it, which insofar as has been demonstrated is all based on individual cells with metabolism, genetic inheritance and compartmentalization.

The list of individual theories, different lines of experimental and theoretical research and diverse views on the OoL is extensive and eclectic. It is not the purpose of this article to be a comprehensive review of all of those. Rather, based on discussions that took part in the 1st Interdisciplinary Origin of Life meeting for early-career researchers (IOoL), we present a forward-looking perspective on how discontinued discourses on the OoL can be (re)united in a new mosaic with resolution and meaning. We reflect purposely on individual topics causing the most distressing divisions in OoL research, most of which result from classical separations between disciplines and theories that date to decades ago. We then portray examples of bridges being built between classically opposed views and finish by providing a roadmap for future dialogue and evidence-based research in OoL.

## 2. Classical Divisions in Origin of Life (OoL) Research

### 2.1. Top-Down Versus Bottom-Up: Where To?

Classical OoL research has gained from synthetic or bottom-up approaches, aiming at synthesizing de novo the building blocks of life (and assemblies of them), and top-down or analytic approaches, which start from living systems, deconstruct them into their parts and survey their properties. But while most origins questions (e.g., that of the universe, atoms, continents or social structures) are usually led by ‘their’ disciplines with small contributions from others, the OoL has been peculiar in calling multiple subjects to play major roles.

Chemistry focused on the heavy lifting at the bottom-up, synthetic side of OoL research. Chemical reactions potentially relevant to life’s beginnings date back as far as 1828 with Wöhler’s urea synthesis [8], a landmark for the birth of organic chemistry. The year of 1861 brought Butlerow’s formose reaction [9], which generates a mixture of carbohydrates from formaldehyde through an autocatalytic mechanism that came to be appreciated by the OoL community nearly a century later [10]. The launch of the chemical origins field occurred with Miller’s experiment in 1953, based on Oparin and Haldane’s theoretical work (see “Section 3. Building Bridges”), synthesizing a mixture of organic compounds (including essential amino acids) from water, methane, ammonia and hydrogen using electric discharges [11,12]. Later, significant advances were made with the synthesis of phospholipid membranes under prebiotic conditions [13]. The synthetic quest continued with efforts focusing on the synthesis of amino acids (usually involving the Strecker or the Bücherer−Bergs synthesis), sugars and nucleobases (usually based on the formose reaction and the chemistry of nitriles and their derivatives); for thorough recent reviews see [14,15] and references therein. Progress is scarcer in understanding the polymerization of proto-biological monomers into oligomers. Recent achievements include RNA synthesis assisted by ribozymes [16,17] and lipids [18]; polymerization of carbohydrate monomers [19] and peptide-bond formation at the water-air interface [20], under dehydration-hydration conditions [21] and in mineral interlayers [22]. Synthetic biology worked on the assembly of more complex features, as the synthesis of a whole bacterial genome [23] and of a viral cDNA genome that was transcribed, translated and replicated without cells [24].

Given that virtually all of the origins hypotheses are non-falsifiable, the diversity of the bottom-up approaches mentioned here has led to an important question: to what extent do the experimental synthetic advances resemble what actually happened at the OoL? With the complete set of inorganics and ready-synthesized organics at its disposal (plus a wide range of experimental conditions), the bottom-up approach parallels an artistic endeavour that can paint both abstract and realistic pictures of the first biomolecules and their assemblies. For this reason, to constrain the experimental space, clearer pictures of i) the fundamental features of life, ii) ancestral life forms and iii) the environmental conditions at their origin are urgently required.

Surprisingly, biology has played a modest part in the search for the OoL [14,25]. A closer look reveals possible reasons for this (quantitatively) sparse involvement. Biology is complex and its numbers are vast: the queue of living species is long, and only recently did technology to start exploring their workings within new disciplines (e.g., biochemistry, molecular biology and genetics) become available. Looking at microbes only, recent estimates point to 1 trillion different species on Earth [26] and orders of magnitude more bacterial cells on the planet than the number of stars in the known universe. A single one of those cells can have at any given moment 2–4 million expressed proteins [27]. By looking at whole living systems, biology has much in its hands, and a gap still remains between i) experimental biology looking at specific mechanistic aspects in a cell or organism (e.g., metabolism, particular enzymes or pathways) and ii) the larger (yet microscopic) picture of how life as a whole started, which falls out of the traditional interpretation of the theory of evolution [28]. Were there mechanisms that drove variation and selection before genomes?

One could say that the top-down approach kick-started when a small number of biological species were compared, the mechanism of speciation was uncovered and the theory of evolution formulated, in parallel with the discarding of spontaneous generation [29]. Yet, the bona fide dawn of top-down approaches in the OoL question would not come to be until the genomics era. Modelling early evolution and the OoL required a precise and holistic way to trace species back in time, and that came first in the form of genome sequences. Prokaryotes, the simplest forms of cellular life, are increasingly supported by evolutionary studies as the oldest lifeforms, and thus of utmost importance for OoL research [30]. The first comparison of prokaryotic genomes revealed a conserved set of 240 genes [31]. Later, the exponential growth of the number of sequenced genomes came with a daunting realization: the prokaryote world is highly diverse, full of redundancy with non-orthologous gene displacements (unrelated sequences encoding the same biochemical function) and lateral gene transfer [32], and the intersection of genomes shrank to a mere handful of ribosomal genes [33]. The search for the genetic content of LUCA—the Last Universal Common Ancestor—using the classical comparative top-down approach had thus stagnated. Some argue that the gap between the OoL and the biochemical wiring of LUCA is so vast that whatever constituted the chemistry leading to the earliest life could have been rewritten multiple times, and changed beyond recognition to then become LUCA [34]. Others argue that even if a good reconstruction of LUCA cannot determine the chemistry at the OoL, universal biochemical features can meet with OoL chemistry by advancing experimental and analytical methods [35]. Parallel but relevant to this line of thought, a new biological concept has been devised—the IDA (initial Darwinian ancestor) [36] also called FUCA (first universal common ancestor) [37], the first entity that could be considered capable of evolution.

The bridging of all the critical advances made by chemistry’s bottom-up and biology’s top-down approaches seems to be an indelible requirement for the advancement of the OoL field [38,39,40]. It is not impossible to envision these seemingly non-overlapping approaches to be reconciled by incorporating expertise from other fields. Indeed, disciplines beyond classical biology and chemistry have much to contribute. Before exploring those contributions, we elaborate further on disputes in OoL research by briefly focusing on another particular format of OoL theories: the prebiotic ‘worlds’.

### 2.2. One Origin, Abundant Worlds

The relative abundance and distribution of building blocks of the main classes of biomolecules ~4 Ga ago on Earth is still elusive and a matter of strong debate [15]. Based on earlier suggestions [41], RNA emerged among all biomolecules in a privileged metaphor that would cement and permeate OoL research for decades to come—the ‘RNA world’ [42]. Ribozymes constituted a shocking and promising discovery: there was now genetic material that was catalytic as well [43,44]. Many found the metaphor appealing: a world with a jack-of-all-trades RNA molecule, catalyzing the formation of indispensable cellular scaffolds, from which somehow then cells emerged [45,46]. Others were quick to notice several difficulties with that scenario. These included the lack of templates enabling the polymerization of RNA in the prebiotic complex mixture [47] and RNA’s extreme lability at moderate to high temperatures and susceptibility to base-catalyzed hydrolysis [48]. The ‘metabolism-first’ theory emerged independently [49,50,51,52], standing in favour of simpler molecular networks harnessing energy from geological disequilibrium leading to the emergence of genetic complexity. The RNA world and the metabolism-first theories have often been portrayed in stark opposition to each other [53,54], leading the OoL field to an unprecedented division. Ever since, other prebiotic worlds came to light with their own preferred class of biomolecules and significant insights, e.g., the protein (or peptide) [39,55,56], lipid [57,58], coenzyme [59], and even virus [60] worlds are some of the most popular theories for the order and/or relevance of appearance of biomolecules on Earth.

Each prebiotic world generated invaluable insight on its own class of biomolecules, but also proposes a privileged, precursory function for them. This privileged position imposes a sequential, stepwise hierarchy on the OoL timeline: subsequent takeovers by the next world(s) until a living cell originates. Still, the question of which class of biomolecules initiated the OoL is a loaded question [61]. All known living cells contain DNA, RNA, proteins, lipids, coenzymes, and other metabolites—and the earliest cells as those known on Earth would have had to fulfil these minimal cell requirements [62]. There is a strong argument to be made for the emergence of essential biomolecules to have been (at least to some extent) contemporaneous and interdependent. More importantly, the origin of biomolecules needs to be distinguished from the origin of cells, and life. Cells are not mere collections of their chemical components, but highly dynamic, complex systems with multiple interlocked processes involving those components. For this reason, the emergence of life cannot be distilled down to biomolecular retrosynthesis only. This appeal is not new, Kamminga asked biochemists to focus on processes rather than on pure retrosynthesis more than 30 years ago [54]. If common requirements between the origin of each biomolecule are found, they may provide necessary constraints for the OoL problem. Ultimately, for the emergence of biomolecules to be relevant for the OoL, it should be tightly linked with their participation in dynamic processes characteristic of life (e.g., replication, energy coupling and compartmentalization). These are some of the reasons why bridging different prebiotic worlds, with all their conceptual advances, is an urgent endeavour.

## 3. Building Bridges

### 3.1. Pressing Questions in OoL are Interdisciplinary

Insights from classical approaches, hypotheses and ‘worlds’ have led to many advances, but they have also resulted in an ideological isolation that is possibly hindering progress in the OoL field [35,53,63,64,65,66,67]. A need to link different disciplines and approaches becomes evident looking at, in our view, the most central questions that should be addressed (Figure 1). These and other questions were identified before [68]. Only cooperation can push their answers forward—in both bottom-up and top-down directions. This is truly not an innovative line of thought. Aleksandr Oparin formulated it in 1924 in a comprehensive way:
*“A whole army of biologists is studying the structure and organization of living matter, while a no less number of physicists and chemists are daily revealing to us new properties of dead things. Like two parties of workers boring from the two opposite ends of a tunnel, they are working towards the same goal. The work has already gone a long way and very, very soon the last barriers between the living and the dead will crumble under the attack of**patient and powerful scientific thought.”*[69]

Both Oparin and Haldane proposed independently integrative theories for the OoL in the 1920s [70]. Their ‘Heterotrophic origin of life theory’ was one of the first approaches to describe a slow transition between a non-living primordial ‘soup’ (Haldane) and living cells. Oparin was the first to integrate data from geochemistry, chemistry, planetary sciences and biochemistry in his seminal work proposing different prebiotic and biological stages [54,69]. His words, quoted above, now almost 100 years old, were optimistic. The “very, very soon” has not happened to this day—there is still a pressing need to bridge (or, in Oparin’s words “tunnel”) top-down and bottom-up approaches. Every attempt to reach over to “the opposite end of the tunnel” opens ways to new data and/or new data interpretations. Cross-disciplinary collaborations can lead to innovative discoveries when new information from discipline A challenges the status quo in discipline B [71]. This forces researchers to formulate new approaches and leads to progress. ‘Outsider information’ like that may help to find a middle ground for irreconcilable hypotheses or theories (see also “Section 4. Towards the Future”) [72].

The OoL question benefits greatly from crossovers of scientific disciplines, each of which brings its specific skills to the table. Chemistry supplies the knowledge of the construction kit, and physics, for instance, the energetic boundaries for the assembly of life [73,74]. Biology is confronted with bigger pictures, providing the roadmap from extant life to its origins via data collection and analysis (from multiple ‘omics’ technologies [75] to the study of elaborate microbial communities [76]). Geology and other geosciences help unfold the possible timeline and set the environmental constraints for the OoL (terrestrial and/or extraterrestrial) [77,78]. All of the above disciplines rely heavily on engineering, mathematics and informatics. Finally, philosophy provides a structure that allows scientists to ask the right questions following the rules of reasoning and argumentation, avoiding the traps of non-sequiturs and other logical fallacies [2,49].

An important first step in connecting disciplines is making the borders between them fully permeable. In the past 50 years, several interdisciplinary fields developed and paved the way for new research. For OoL in particular, astrochemistry, astrobiology and their search for biosignatures in the universe had a significant impact. These disciplines focused on physical and chemical processes that could lead to biotic or prebiotic systems on other planets [79] (Figure 1). One example is the discovery of subsurface oceans on icy moons, which extended the concept of habitable zone from the inner to the darker, colder places of the outer solar system [80]. Both complex, large insoluble organic material and low mass, reactive and soluble molecules, have been found in the subsurface ocean of Enceladus [81,82]. Large molecules have also been found in the atmosphere of Titan [83].

How does the need for discipline crossovers and permeability translate into practice? To illustrate this, we will use three central questions—all concerning essential cellular processes—as examples: (i) how did energy coupling arise, (ii) what mechanisms led to metabolism, and (iii) how did the genetic code emerge?

Life on Earth couples energy-releasing (spontaneous) reactions to energy-demanding (non-spontaneous) ones, capturing energy from its environment and eventually dissipating it as heat. This enables cellular processes such as growth and division. But how did this sophisticated system develop? Today, energy-coupling is mediated by enzymes which, acting as engines, funnel energy released from the cell’s diet into chemical energy. This energy is stored in a thioester linkage (as in acetyl-CoA), a phosphate-ester bond to carbon like in acetyl phosphate or a phosphate bond in the adenosine triphosphate (ATP) molecule [84]. These molecules are commonly known as energetic currencies in cells and mediate energy coupling by transferring energy between non-related biochemical processes. Are energetic currencies indispensable or exchangeable within existing metabolisms? To answer this biological insight on different species and pathways is crucial [85]. In order to find out if energy coupling was necessary for life’s first steps (still a subject of debate) several alternative energy currencies have been proposed and tested, for instance high-energy phosphates [86] or thioesters [87]—as it is known that cells use both, often sequentially (e.g., the production of ATP is sometimes mediated by thioesters). From there, further questions emerge: how did these energy-storing molecules become coupled and implemented into life as we know it (or: what was the order of entry of such heteroatoms as nitrogen, sulfur and phosphorus into proto-life’s chemistry)? Here, it would be helpful to look at different geochemical conditions to explore under which conditions these couplings could be possible. Although the answers are still unclear [88], discovering them will be pivotal to understanding the transition between non-living and living systems [84,89].

The question of energy coupling is directly linked to the question of what geochemical mechanisms led to complex cellular metabolism. A systematic search for metabolism’s beginnings (i.e. protometabolism) requires a clear and agreed-upon definition of metabolism, yet to be developed. This definition should start from the molecular level and the fundamental physical driving forces (kinetics vs. thermodynamics) [90], leading to combinations of chemical reactions, ultimately giving rise to a complex network [91,92], governed by a set of rules and boundaries [93,94]. Starting from known biochemical pathways one can deduce a conserved network that could sustain itself without enzymes [95,96,97]. The feasibility of such top-down analysis reducing the complexity of biological metabolic networks has to be tested in a bottom-up laboratory setting [98,99,100]. Conditions for these laboratory experiments can either be inspired by known geological environments, or a set of favourable conditions can be searched among existing geological settings and/or models. Such comparisons and convergences between biochemical and geochemical conditions, reactions and products can help to unveil environments where metabolic pathways and networks could emerge [101,102,103]. This applies not only to the early Earth but also other locations elsewhere in the universe like Enceladus [81,104]. Certainly, the same methodological rules will apply to other angles from which the OoL question is tackled: researching compartmentalisation, reconstructing the characteristics of LUCA, studying molecular replicators and precursors to genetics —hardly can we envision any of these succeeding in isolation.

Along with metabolism, life is based on another equally-important fundamental principle-inheritance, also described as “information that copies itself“ [105]. As discussed (see “Section 2.2. One Origin, Abundant Worlds”), a better question than “which came first, information or metabolism?” would be: does the emergence of one have common properties with the other? In other words, perhaps we will know that acetate originated before ATP, but the emergence of metabolism and inherited information as complex systems was most likely interdependent and simultaneous. This directly leads to the fundamental question of how and when metabolism and information storage became linked. Nature’s elegant solution is the genetic code, the origin of which remains a true enigma.

Different hypotheses exist for the origin of the genetic code, often depending on assumptions of the different hotly debated prebiotic conditions. Ribozymes show that RNA can harbour both catalysis and genetics, and there is support for the capability of RNAs to aminoacylate [106]. However, the code today is self-referential, that is, the mapping between amino acids and codons heavily relies on encoded proteins. When, why and how did non-coded peptides become involved? For the origin of the involvement of peptides in the genetic code, the conjecture that they are a superior catalyst is not enough, especially because they only have a supporting but not catalytic role in the ribosome [107]. Therefore, we must justify likely proximal selective advantages of using amino acids and simple peptides in the context of RNAs. These so-called ‘exaptations’ may bridge the RNA and peptide worlds, offering the basis on which later elements of the translational apparatus were built. Possible advantages of peptides include providing catalytic aid and expanding the catalytic repertoire of RNAs [107], membrane transport [108], scaffolding [109] and energy storage [110]. It has indeed been shown experimentally that non-coded peptides can potentiate the functions of RNA, which supports the coevolution of RNA and peptides [111]. Both the RNA and protein worlds ask whether a bipartite polymer setup (with nucleic acids and proteins) and the current genetic code is an inherent requirement of life, or if it is possible to envision one dominant polymer carrying information in prebiotic stages. Neither world alone can provide a clear explanation for the interlacing of the two, though, and there are relatively few endeavours to try out ‘messy emergence’, where some initial cooperation between non-coded proteins and RNAs was vital. However, regardless of the polymers at the origin of the code, another question urges: how did the code freeze to the current codon table? This question seemed for some time to be a mere statistical or even cryptographic problem [112], and a variety of explanations emerged to solve it, most popularly: stereochemical basis for the assignment between nucleic acids and amino acids [113], the development of the code guided by the biosynthetic pathways of amino acids [114], and optimization in order to reduce the severity of mutations [115]. These are not mutually exclusive hypotheses, and the origin of the code might have simultaneously involved several of them [116]. Most likely, the development of the genetic code took place in a continuous expansion [117], a hypothesis supported by functional proteins with reduced amino acid repertoires [118]. The later expansion of the repertoire was possibly governed by a physicochemical optimization that fits well with water-based biochemistry [119]. 

Answers on the origin of the code still seem very distant [115,116,120,121,122]. In order to finally solve this puzzle it will be necessary to investigate not only the emergence of biological information [120], but also the evolution of interactions between the molecules involved in translation [117,123].

### 3.2. On the Right Track? Looking at the Past Decade

In the past 10 years, many have worked and asked for OoL research to unite [25,39,124,125]. Here we look at experimental examples from the last decade that connect different disciplines, theories or interpretations (Figure 2).

One of the first barriers to come down stood between the decades-old views of different single-biomolecule worlds (see “Section 2.2. One Origin, Abundant Worlds”). Studies merging the lipid world with others are pioneers. The requirement for compartmentalization to keep genomic molecules and their products spatially together, as well as to allow for vectorial (bio)chemistry, suggests that the potential of lipids and other amphiphilic structures to self-assemble into micelles and bilayer vesicles within an aqueous phase constituted a critical step in the emergence of life [57,126,127]. Vesicles were thought to be stable only in salt-poor aqueous environments, such as surficial freshwater ponds or hydrothermal springs, but recent work showed that these become much more resilient to extreme salinity and pH if they are composed of mixtures of amphiphiles [128], a feature which better reflects the naturally messy aspect of prebiotic chemistry. Such prebiotic compartments, also referred to as ‘protocells’, are defined as primitive precursors of modern cells which, although not yet alive, exhibited essential cellular characteristics [129]. Efforts of OoL researchers aim to establish various in vitro protocell models that mimic key features of life as we know it, including simple metabolic pathways, replication or vesicle growth and division [130]. A protometabolism leading to sugar synthesis has been assembled within lipid vesicles, with the final products diffusing through the lipid barrier and being detected by living bacteria [131]. Not long after, DNA amplification was shown to induce growth and division of lipid vesicles, linking the reproduction of an informational substance with that of a compartment [132]. These demonstrate that simpler systems than cells can model fundamental interactions between membranes and their contents. Another example is the bridge between the metabolism-first and the RNA world theories [114]. It is now clear that the building blocks for RNA and DNA are intermediates of metabolic networks; they are never directly uptaken from the environment in their ready-for-polymerization forms (i.e. as phosphorylated nucleosides), but as unphosphorylated biogenic nucleosides [133], and are also in fact moieties of several essential cofactors [134]. The idea of the simultaneous and interdependent origins for RNA and DNA’s building blocks has been given experimental evidence [135]. But bridges between the RNA world and metabolism-first can be built beyond their typical molecules. Classical approaches in OoL have often been constrained to biomolecules due to their ubiquity in biology, however, most of these biomolecules were not necessarily available at early prebiotic stages [136]. Prebiotic environments most likely included compounds not central to modern biopolymers, for example, alpha hydroxy acids (aHA) [137] among numerous others [138,139]. Recent work has shown that aHAs easily form combinatorial polyesters under wet-dry conditions that may have played a role in the catalytic landscape within which they were formed [140]. These polyesters can form membraneless compartments mimicking a cell, capable of differentially segregating various kinds of dyes, hosting a protein and even accumulating lipids in their exterior [141]. If one considers that the accumulation of biomolecules is a biological invention, optimizing for certain geochemical and/or biochemical properties [119], then compounds such as aHAs may have played a role in some form of nascent biology not conserved in modern biology [142]. Considering the ‘messy’ nature (meaning the inherent diversity) of prebiotic chemistry [143,144], this may have been the case, and the utility of central biomolecules such as RNA and ubiquitous metabolites to elucidate the OoL may be partial, despite their omnipresence in modern life.

Other striking examples for integration in OoL research come from geological studies providing a better picture of the Hadean world, including the likely atmospheric composition and the depths of oceans and tectonic activity [145,146]. These studies constrained and brought closer the work of both biologists and chemists. Experimentation with origins in hydrothermal vents [147,148,149] and geothermal fields [150,151,152] have taken into consideration geological insights in their OoL scenarios. In particular, the importance of relevant metals and metal clusters in the Hadean has settled in experimental work that recreates the origin of biochemistry in vitro, including carbon cycling [98,103,153], nitrogen fixation [154], ribosomal translation [155], and even the generation of pH gradients [156]. Simulated hydrothermal conditions, in particular pores, select for the replication of longer oligonucleotides [157]. Mineral surfaces have shown promising features in promoting biochemistry, including the selection of longer RNA molecules [100], and a variety of organic reactions including nitrogen reduction, lipid self-organization, condensation-polymerization reactions, selection and concentration of amino acids and sugars and chiral selection (see [158] and references therein). Surfaces can also help to tame the combinatorial space in a complex system of organic molecules [100,159].

By providing new ways to handle vast amounts of data in a systematic and quantitative manner, advances in computer sciences and technologies help the much-required aforementioned integrations that are starting to reflect on OoL research. Larger chemical libraries can now be monitored over a manageable timeline, and the diversification of self-replicating molecules has been observed in such systems [160]. The self-replication of small organic molecules has also been observed in an autocatalytic process displaying complex, non-linear responses to changes in environmental conditions [161]. Computational models allow us to test hypotheses unattainable in a laboratory alone, be it chemical conditions [22], timescales [4] or other routine perturbations to the model. Computational biology has now moved to mathematical simulations at increasing levels of complexity, from topological, interaction-based to constraint- and mechanism-based [162]. Recently, a database aggregating a variety of bioinformatic approaches to LUCA was compiled, allowing for testing hypotheses in investigations of ancient biochemistry [163]. Simulations can now be undertaken with geology and biology in mind, both in molecular dynamics [22] and network biology [95,96], identifying constraints and parameters that can provide direction for experimental work [98,159,164].

The progress of top-down approaches from traditional comparative genomics to more integrative approaches revives optimism in the quest for LUCA [165]. Integrating fossil data with molecular evolutionary clocks has solidified evidence for the age of LUCA at >3.9 Ga [4]. Taking into account morphological characteristics together with genetic signatures is another promising direction. Recent analyses of membranes and cell walls of prokaryotes suggest that LUCA was able to sporulate, that is, to reproduce into a dormant, non-metabolising cell that could survive long periods of harsh conditions such as the late heavy bombardment [146,166]. Modern phylogenetic considerations minimize the effect of lateral gene transfer in reconstructing the first genomes, pointing to a thermophilic autotrophic LUCA [149].

## 4. Towards the Future

### 4.1. General Remarks

We expect by now to have presented a convincing argument for the necessity of bridges in answering OoL’s most prominent questions (Figure 1), together with examples of successful studies (Figure 2). How can the future hold more interconnections between approaches and theories? Some tools are general for all types of linking between seemingly disconnected research (Box 1). These include interdisciplinary conferences, where the focus lies not only on one molecule or one theory, but either on the whole field or on the question (Figure 1) sought to be answered. Interdisciplinary higher education and scientific collaborations should progress in the same direction in the coming decades, with more focus on questions and holistic pictures.

Box 1Tools for future integration of OoL research.
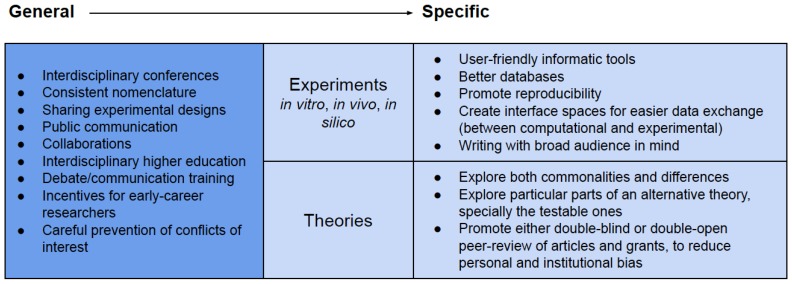


Regarding the limits of experimentation, computational tools can extrapolate from experimental input, and thus will play a major role in untangling complexity in OoL. However, research in vitro, in vivo or in silico (Box 1) needs to be attainable to everyone. This requires more user-friendly computational tools, the communication of tools and results in an accessible manner, and the creation of curated and moderated spaces where data and tools can be exchanged, faster than by the traditional scientific paper (as forums, curated databases, websites and discussion channels). Standards that can be used by different disciplines are heavily required, particularly in chemical, biochemical and genetic nomenclature—we recommend Chebi [167], KEGG [168] or BiGG [169]. Nomenclature in theoretical biology also requires attention. In particular, nomenclature regarding the first biological entities includes a great amount of confusing variability, e.g., an initial Darwinian ancestor (IDA) [36] akin to a first universal common ancestor, FUCA [37], and the more commonly used LUCA, akin to the urancestor [170], to the cenancestor [171], and to the (simpler) universal common ancestor [172]. Here, we recommend, merely based on the greater usage to this day, the use of IDA and LUCA. Semantic variability without clear conceptual justification does not ease literature review and theoretical studies, and a strong community effort is required to make different names clearly distinguishable or for standards to be adopted.

Sharing and combining experimental setups (both computational and laboratorial), can help to increase reproducibility but also to connect disciplines and approaches. For example, to narrow the current gap between prebiotic (geo)chemistry and ancient biochemistry, simulation efforts could focus on the least contested facts on the early Earth. These include a CO_2_-rich atmosphere and hydrosphere from volcanism, and an H_2_-rich hydrothermal outflow from the crust. Other parameters which are harder to characterize, as pH ranges, pressure, access to ultraviolet (UV) radiation, water activity, etc. could then be varied. There are means to attempt such simulations in the laboratory today, where collecting vast amounts of data is possible [173]. This data can be analysed and incorporated into the computational models which then, in turn, can shed more light on what laboratory experiments should be attempted. Hereby, the steady bilateral exchange between disciplines is crucial. Not only should geology provide conditions for experiments [174], those experiments should also provide the attributes that geochemical studies could search for. 

### 4.2. Commonalities between Opposing Theories

Bridges between methodologies will allow the OoL field to advance, but ultimately theories will have to become more interconnected as well. We foresee that this can happen with an intentional focus on commonalities rather than solely on differences, and plenty of examples can be mentioned. Here we choose to focus on four of the most heated debates, heavily interrelated: (i) the geological setting for the OoL, (ii) the source of food/starting molecules, (iii) the source of energy and (iv) the RNA world vs. metabolism-first division.

#### 4.2.1. The Geological Setting

The proposed and heavily discussed locations for the OoL, including terrestrial hot springs [150], submarine hydrothermal vents [175,176], volcanic landscapes [177], hydrothermally-percolated sediments [178] and warm little ponds [179] create some of the strongest disagreements in OoL research, but have in common several requirements. These include a constant energy and CHNOPS source, heterogeneous or homogeneous catalysts, non-equilibrium conditions and conditions that enable aggregation of molecules to form life’s building blocks and, eventually, polymerization of said building blocks [22,103,152,180]. One example is the requirement for wet–dry cycles for polymerization, mostly discussed in the context of geothermal fields [180], even though wet-dry cycles can also be found in hydrothermal pores [181] and even at the water-air interface [20]. On another note, some propose that different geological settings could have created different biomolecules that were then able to meet in an unchanged chemical state in the location where cells ultimately emerged [182].

#### 4.2.2. The Food Source

A historically important division drawing from the analogy with different types of metabolism in prokaryotes [125] is the heterotrophy vs. autotrophy conundrum. Each side has envisioned their abiogenic scenario from a trophic point of view: if the carbon source for organosynthesis was eminently inorganic (usually CO_2_), this represented an autotrophic origin (e.g., [183]), whereas if the carbon source were reduced organic compounds, a heterotrophic one (e.g., [184]). This division usually links to whether the earliest cells are considered autotrophic (e.g., [149]) or heterotrophic (e.g., [185]). It is important to note that for the first cells, the question is indeed meaningful—those cells either imported C_1_ compounds, subsequently reducing them and forming C–C bonds (autotrophy) or imported C_n_ compounds and used those directly in biosynthesis [186,187]. But others are wary that this division may be unhelpful before the cellular stage (see Smith and Morowitz in [68]), since trophic types apply poorly to biosynthetic pathways. All organic molecules synthesised non-enzymatically ultimately derive from an inorganic carbon source (C, CO or CO_2_) which was subsequently reduced. The endogenous (i.e. terrestrial origin) vs. exogenous (i.e. extraterrestrial origin) debate also entails limitations, by simply referring to the physical origin of the organic molecules, but not informing on the chemical relationship between these and the nascent proto-biochemistry. For the origin of the first biochemical networks, potentially acellular IDAs, we propose referring instead to whether all or some of the reactions leading to organic molecules occurred in situ (i.e. within, or physically contiguous to the nascent proto-biochemistry) or ex situ (i.e. physically separated from it). Note that this is not just a geographical distinction. Instead, it aims to find out whether the nascent chemical network (eventually leading to living cells) could affect (positively and/or negatively) the chemical reactions feeding it. This, we believe, is a meaningful change in language aiming to mend the limitations associated with the autotrophic vs. heterotrophic dilemma. Therefore, the in situ synthesis implies the reactions yielding biomolecules as part of the nascent protometabolism. In contrast, in the ex situ subtype the synthetic reactions rest unaffected by the nascent chemical network, and the latter needs to invent alternative chemical pathways to eventually become independent from the former’s unidirectional supply.

#### 4.2.3. The Energy Source

Non-equilibrium conditions suggest an important bridge that can be established between most OoL hypotheses [14,188]. The fact that life is a process that is by definition not in equilibrium is not debatable in physics, biology or any other of the involved disciplines [189,190]. One example of how ‘disequilibration’ can be observed in life as we know it, is that life builds up complexity and then breaks it down again—in biology, this is, in broad strokes, anabolism and catabolism. These complex processes in general parallel energy spending and gaining in the cell, respectively. However, some organisms (e.g., methanogens) are known to conserve energy only through anabolism [191]. The earliest life would have had to couple both the constructive and destructive regime in some way [188,192]. To fulfil their energetic requirements, prebiotic systems would need both electron donors and acceptors supplied by their environment. The source of energy for the earliest life has been strongly disputed, with plausible hypotheses ranging from pH and redox gradients to thermal energy and UV light (for a detailed discussion see [74,193,194]. It is, however, not unlikely that several energy sources played a role in different prebiotic stages. Nevertheless, other constraints than the primary energy sources are imposed as soon as more complex prebiotic systems arise. A cell-like energy-coupling system could only be persistent over time if it can be inherited, thus forging a necessary link between the offspring hypotheses of the classical ‘metabolism-first’ and ‘genetic-first’ worlds [194].

#### 4.2.4. RNA World versus Metabolism-First

Modern thinking on the OoL highlights the need for not only the synthesis of life’s building blocks (themselves crucial goals), but also the reenactment of the processes they participate in [195]. Thankfully, what follows from this realization is that metabolism- and genetics-oriented hypotheses are gradually ceasing to be seen as mutually exclusive, because cellular processes involve both metabolites and genetic molecules [25,124]. A particular example regards evolution. Looking for the origin of natural selection and evolution is to look for the origin of an inheritance system that must maintain the capacity to produce new combinations, while keeping fidelity in information transfer [196,197]. But what *is* this information? Biological information is more than a collection of bits; it has contextuality, translated into functionality and is prone to evolve, as languages do [198]. In other words, genetic molecules only hold information in the right context: when they can be recognized and translated to functions that keep the system going. This definition opens different avenues to look for information storage and transmission during chemical evolution. 

The earliest information could have been made from the types and quantities of molecules within a self-sustaining chemical assembly, also called ‘composome’ [199] or autocatalytic network [200]. This relates to the emphasis by some recent OoL research on the co-dependence of metabolism, containment and replication/information (see “Section 3.2. On the Right Track: Looking at the Last Decade”). This emphasis is central to bridging the RNA world and metabolism-first theories and points to the origin of molecular coordination or cooperation. Following this line of thought, some authors propose early ‘selfish cooperators’, akin to viruses, which formed stable ensembles of co-inherited genetic elements [60]. These ensembles should have been able to perform both a kind of proto-replication and a kind of proto-metabolism [201]. Still, there is a ‘cooperation barrier’ in the transition of non-life to life, caused by (i) molecules that could cooperatively contribute to the success of an ensemble but which are often not supported by the ensemble, and (ii) side reactions or processes that undermine cooperation [202]. To overcome this barrier, the management of those otherwise unconstrained ensembles is required [202]. This hypothesis is particularly interesting because it exposes the necessity of a digitally-coded (genetic, on/off) management of the analog (continuous) reactions of metabolism in order to overcome the cooperation barrier efficiently, reflecting the two-tiered structure of all known living cells [198,202].

If such analog information storage mechanisms as autocatalytic networks are able to undergo Darwinian evolution is a matter under debate [93,203]. Several have suggested fitness criteria for natural chemical selection, including the rate of entropy production [204] and other kinetic or thermodynamic features of chemical reactions, such as stoichiometric catalysis, autocatalysis, and cooperativity [205]. A specific example is the greater reactivity of proteinaceous amino acids when compared to their non-proteinaceous counterparts, which naturally selects from oligomers of the former [206]. At some point, most likely quite early during the development of prebiotic systems, genetic molecules did crystallize, and information storage became based on molecular recognition. Every letter of the ‘genetic alphabet’, A, C, G, T and U, given a recognition/translation system, contains information. Early molecular recognition could have been significantly different to contemporary DNA and RNA [207,208] and even minerals have been proposed as information storage molecules [209]. 

What we know now, is that if we wish to search for answers on the origin of life as we know it—meaning the origin of cells—we are faced with an increasing conceptual [2] and experimental complexity [210] that will require the integration of RNA molecules with peptides, lipids, and protometabolism. Better models of complex biological systems [100], as well as better techniques to characterise them [211], will help to tackle such complexity. However, this complexity can no longer be denied. In other words, both the RNA-world and metabolism-first theories aim at the origin of life as we know it. However, we do not know it without each and both of them.

## 5. Conclusions

The main reason the OoL research field is still divided on so many issues is that it seems virtually impossible to find definite answers for all of our questions—we cannot wait hundreds of thousands of years to observe in real-time if a certain geological setting trumps another or which are the first enzymes to develop. We can only approach solutions asymptotically and, returning to the image of an OoL mosaic, add one pebble at a time, one insight that brings us closer to a more complete picture. In this article, we tried to convey how this is already being done—and hopefully, can be done even better in the future. Just as life itself is a synergistic process, in the sense that life’s biological, physical and chemical properties are intertwined, the research that tries to explain its origin has to be synergistic as well. No discipline or approach should be dismissed, as long as its claims are evidence-based. However, we should keep in mind that while single results have dominated the field in the past, these can never constitute a solution to such a complex problem. Synthesising biomolecules, for example, although informative, cannot be enough to tackle the OoL, because life is not a mere collection of biomolecules, but rather a dynamic process involving biomolecules within boundary conditions interacting with its environment. Modern thinking on origins will thrive from recreating or imitating processes, rather than focusing only on the prebiotic synthesis of biomolecules. 

Finally, we must highlight bias in OoL experiments, starting from how to define ‘plausible’ prebiotic conditions to which biomolecules are more central than others [212]. It is imperative to overcome personal biases in order to make progress. To get there, we need to increase the exchange, openness and respect between all those involved despite the inherent competition and inherited bias towards different hypotheses or approaches in research these days. The bridges we propose are essential, not because scientific research must be homogeneous, but because heterogeneity must be articulated (for examples see “Section 3. Building Bridges”). This articulation will change both the way research is done but also communication (in published articles, conferences and others). At this point, it is important to emphasize how crucial classical approaches in OoL research were and still are. However, while pivotal insights were achieved in classical studies, a more complete OoL narrative can only unfold by building bridges between them. This might all appear far too obvious to mention, but the reality in origin of life research shows that it cannot be mentioned often enough.

## Figures and Tables

**Figure 1 life-10-00020-f001:**
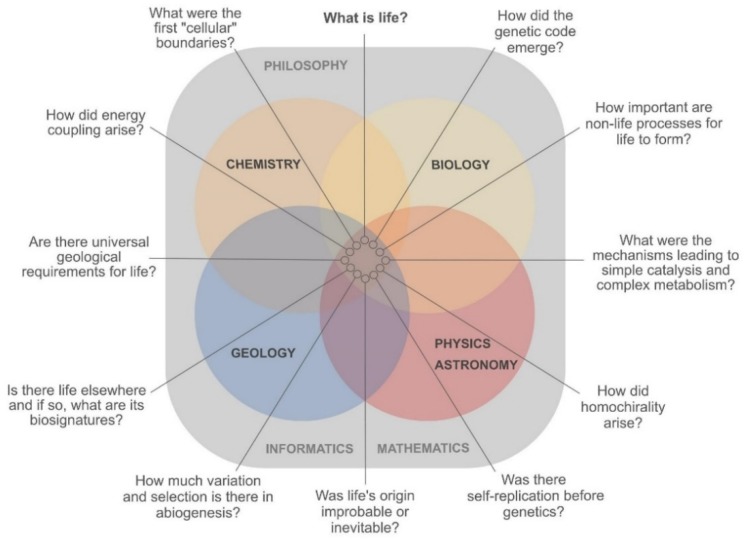
Current central questions in origin of life (OoL) research. In order to find answers, each of the natural sciences illustrated has to play a role by converting these questions into hypotheses and theories, while constantly testing them experimentally. The fundamental roles of philosophy, mathematics and informatics are portrayed in the background.

**Figure 2 life-10-00020-f002:**
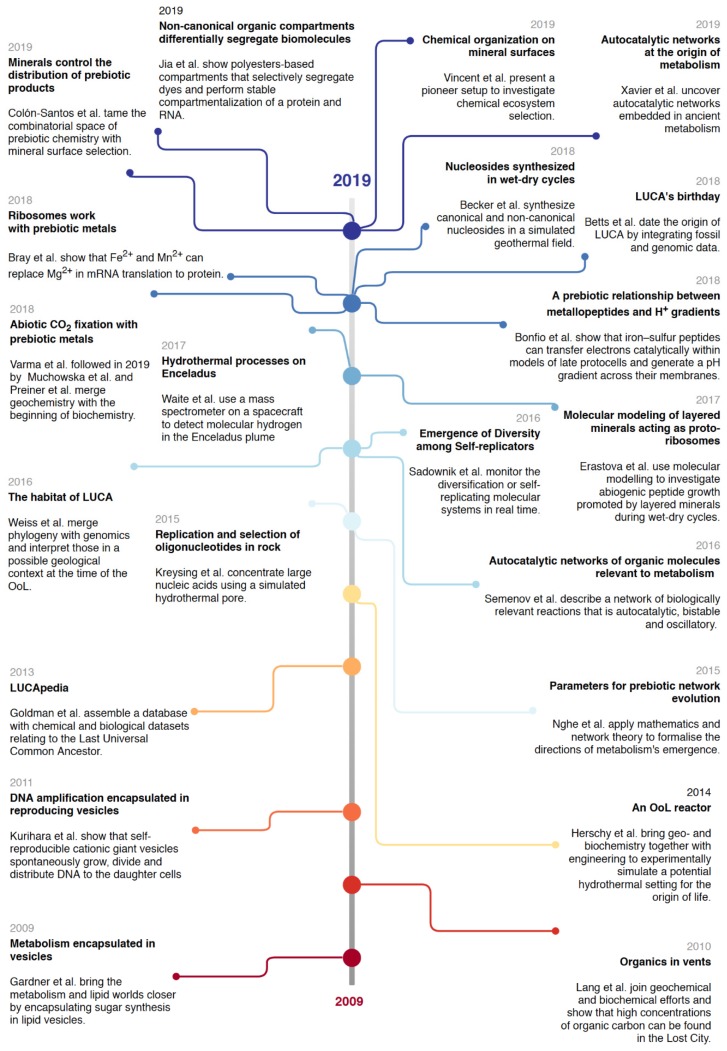
Timeline of recent multidisciplinary achievements that build bridges in OoL research. Included are examples from the past 10 years of OoL research that bridge disciplines, approaches and/or methods, biomolecules/single-world scenarios, simulations and experiments. The choice of studies does not aim to cover (exclusively) novel findings, but those that build bridges.

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
