# Peer review of "The Future of Origin of Life Research: Bridging Decades-Old Divisions"

_life, 2020, doi:10.3390/life10030020_

Round 1
Reviewer 1 Report
The revised manuscript looks good.
Author Response
We thank Reviewer 1 for the positive evaluation of our revised paper.
Reviewer 2 Report
This is good in all the ways I’ve noted before, and much-improved on most of my concerns.
Skimming though responses to other reviewers, I would tend to agree with the authors there, too.
All good stuff!
In fact, there is only one set of issues which remain are within the text Fig 2. This is a 'Perspective', and I do not feel a reviewer should stand in the way of publishing opinions. I do feel that some portion of the text in Fig 2 risks misleading the reader, though, and do not feel comfortable “signing off” on that. I have mentioned this already, so I hope a misunderstanding is responsible for the remaining issue. I will set out in a little more detail below (along with other misleading claims on further inspection), and mark this for “minor correction”.
I look forward to seeing this in print very soon.
Issues with Fig 2 text:
Fig 2 is much easier to read. I appreciate the authors do not intend it to be comprehensive. I do not suggest it should be. Text giving false impression of “firsts” is different to omission, though, and risks misleading the reader.
Below I note ongoing issues with descriptions I had already highlighted. Beginning to dig more, I notice more misleading text, and note a couple of others.
For example, issues I have already highlighted:
- “Erastova et al show that wetting-drying cycles of layered minerals promote abiogenetic peptide growth.” The “show” suggests a first, which would be misleading. I’ve already commented: Lahav et al demonstrated abiogenetic peptide growth by dehydration on layered minerals in the 1970s [Lahav, et al. Science 201, 67–69 (1978)]. This modern work is certainly not the same as Lahav, so I’m sure a less misleading description could be found. (Perhaps the key is in being purely theoretical work?)
- Similarly, the “Jia et al” Fig 2 text still suggests “non-canonical organic compartments” as if this is an innovation. It is not. For example, de Jong’s coacervates, almost 100 years ago. And, of course, compartments like this from amide/ester condensation polymers are not new either (e.g. Fox, since the 1950s). Again, perhaps best to concentrate on what makes this paper distinct/new/important, or pick work which better illustrates recent advances.
Digging further (not comprehensive):
- The “Colon Santos et al” text claims “Automated testing of prebiotic chemical space”, suggesting a “first”. Oddly, looking through the paper, I find no report at all of reaction automation (thought that group has reported prebiotic chem reaction automation in other papers…!?!). There seems to be scripting/automation of data analysis; again, though, this isn’t’ the first paper from that group to automate data analysis. Indeed, the Colon Santos et al paper cites previous examples from that group, including w/mineral surfaces limiting combinatorial space in prebiotic chemistry. Furthermore, other contemporary groups have also published using advanced analysis (Forsythe/Hud). Again, perhaps best to concentrate on what makes this paper distinct/new/important, or pick work which better illustrates recent advances.
- Conversely, the “Bonfio et al” commentary doesn’t mention generating a transmembrane potential, which is odd.
I sympathise with authors wishing to show their/colleagues’ work in a favourable light, but perhaps less misleading descriptions could be found? These examples have jumped out at me, but I’d suggest the authors take critical look at the other descriptions.
Author Response
We thank R2 for the positive comments regarding the revised version of our manuscript. We are particularly pleased when reading his or her view concerning the publication of perspective papers.
Regarding the concerns with Figure 2, we did not, in the previous reply to referees, explicitly quote the changes made. In reflection this was clearly a lapse from our side. We had introduced them, hoping to answer R2’s concerns. Particularly, the previous revision included changes for the descriptions of the papers by Jia et al. 2019, Erastova et al. 2017 and Hershy et al. 2014. We appreciate how the changes could have been insufficient. Please note that we never intended to portray these as “new findings”, or have we suggested so in the text. We call them “examples from the last decade that connect different disciplines, theories or interpretations” and “recent multidisciplinary achievements that build bridges in OoL research”. The claim of novelty is a bold one, which we never intended to do. None of the descriptions says “for the first time” or similar, and the interpretation that we were giving “false impression of firsts” was sincerely never our intention.
Even if an interpretation, we believe that R2’s is possibly a common interpretation. Therefore, we introduced the following changes in this second revision:
Caption of Figure 2: The last sentence was changed to: “The choice of studies does not aim to cover (exclusively) novel findings, but those that do build bridges.” (lines 431-432, in blue in the revised manuscript).
Changes in Figure 2
Jia et al. 2019: changed from “Non-canonical organic compartments; Jia et al. show alpha hydroxy acids assembling in compartments with potential prebiotic cellular functionalities.” to “Non-canonical organic compartments differentially segregate biomolecules; Jia et al. show polyesters-based compartments that selectively segregate dyes and perform stable compartmentalization of a protein and RNA.”
Colón-Santos et al. 2019: changed from “Automated testing of the prebiotic chemistry space; Colón‐Santos et al. tame the combinatorial space of prebiotic chemistry with mineral surface selection.” to “Minerals control the distribution of prebiotic products; Colón‐Santos et al. tame the combinatorial space of prebiotic chemistry with mineral surface selection.”
Erastova et al. 2019: changed from “Peptides grow in layered minerals through wet-dry cycles; Erastova et al. show that wetting-drying cycles of layered minerals promote abiogenic peptide growth.” to “Molecular modelling of layered minerals acting as proto-ribosomes; Erastova et al. use molecular modelling to investigate abiogenic peptide growth promoted by layered minerals during wet-dry cycles.”
Bonfio et al. 2018: added to the description: “and generate a pH gradient across their membranes”.
We thank R2 for helping us to reduce our potential bias, and hope that the revised version is in accordance with scientific standards.
Reviewer 3 Report
I have now understood well the authors’ attitude of tackling OoL research through the replies to my comments. Therefore, I would like to recommend the editor to publish this manuscript in “Life”. I am convinced that the paper will play a significant role in the future OoL research.
Author Response
We thank Reviewer 3 for the positive evaluation of our revised paper and are also grateful for the encouraging words regarding its impact.